# Liver Sinusoidal Endothelial Cells and Their Regulation of Immunology, Collagenization, and Bioreactivity in Fatty Liver: A Narrative Review

**DOI:** 10.3390/ijms26168006

**Published:** 2025-08-19

**Authors:** Reem J. Abdulmajeed, Consolato M. Sergi

**Affiliations:** 1Department of Pathology and Laboratory Medicine, University of Alberta, Edmonton, AB T6G 1B7, Canada; reem_a.m@hotmail.com; 2Division of Anatomic Pathology, Children’s Hospital of Eastern Ontario, University of Ottawa, Ottawa, ON K1H 8L1, Canada

**Keywords:** liver, MAFLD, NAFLD, NASH, MASH, hepatocellular carcinoma

## Abstract

Liver sinusoidal endothelial cells (LSECs) are essential for preserving liver homeostasis. Metabolic dysfunction-associated steatotic liver disease (MASLD) encompasses a category of hepatic disorders characterized by excessive fat accumulation in the liver, known as steatosis. Over time, accumulated hepatic fat can induce inflammation of the liver (hepatitis). MASLD is among the most prevalent types of chronic liver disease. Obesity and Type 2 diabetes mellitus (T2DM) are frequent etiological factors of MASLD. In the absence of therapy, MASLD can lead to more severe hepatic conditions, which can be life-threatening. MASLD is noteworthy due to its potential progression to MASH and further severe liver impairment, including cirrhosis and hepatocellular carcinoma (HCC), a neoplastic progression. This narrative review examines the distinctive functions of LSECs in regulating immunologic responses, collagenization, and drug-sensitive bioreactivity in healthy livers, MASLD, and metabolic dysfunction-associated steatohepatitis (MASH), as well as in a human primary 3D model. We found that LSECs serve as crucial regulators of immunological equilibrium in the liver by inhibiting disproportionate immunologic activation, concurrently filtering tissue antigens, and engaging with immunologic cells, such as Kupffer cells (KCs) and T lymphocytes. In chronic diseases of the liver, LSECs experience cellular dysfunction, resulting in capillarization (focal to diffuse), loss of fenestrations (*fenestrae*), and the activation of pro-fibrotic signaling pathways, including transforming growth factor-beta (TGF-β). Indeed, TGF-β is crucial in activating hepatic stellate cells (HSCs), a process that facilitates the progression of liver disease toward fibrosis. In addition to examining the dynamic interplay between LSECs, specifically HSCs, and other liver cells throughout the progression of fatty liver–MASH, we suggest that LSECs may become a potential therapeutic target for modifying immune responses and averting fibrosis in hepatic disorders. The limitations of animal models are also highlighted and discussed.

## 1. Introduction

Liver sinusoidal endothelial cells (LSECs) are specialized cells that line the hepatic sinusoids, the capillaries that allow blood to flow between the liver and the rest of the body [1,2]. These cells are crucial for the liver’s functions, including filtration, nutrient exchange, and immune regulation. LSECs are also involved in liver diseases, such as non-alcoholic fatty liver disease (NAFLD) (currently labeled as metabolic dysfunction-associated steatotic liver disease (MASLD)), fibrosis, and cirrhosis with completely overlapping scores, as recently identified [3,4,5]. MASLD encompasses a category of hepatic disorders characterized by excessive fat accumulation in the liver, known as steatosis. Over time, accumulated hepatic fat can induce inflammation of the liver (metabolic dysfunction-associated steatohepatitis, MASH). MASLD is among the most prevalent types of chronic liver disease. Obesity and Type 2 diabetes mellitus (T2DM) are frequent etiological factors of MASLD. In the absence of therapy, MASLD can lead to more severe hepatic conditions, which can be life-threatening.

Unique features of LSECs include fenestrations (from the Latin “*Fenestrae*”) or windows (pores that make the cells highly permeable), minimal or no basement membrane (unlike other endothelial cells), and a richness of scavenger receptors that effectively remove various molecules from the bloodstream. Thus, a more detailed examination of LSECs will highlight their functions in the liver, which include filtration and molecular nutrient exchange, immunologic regulation, blood flow regulation, scavenger function, and liver regeneration [1,2,6,7].

Regarding filtration and nutrient exchange, LSECs function as a permeable barrier, allowing the passage of molecules between the blood and liver tissue, thereby facilitating nutrient uptake and waste removal. In immunologic regulation, LSECs play a role in immune responses within the liver, including antigen presentation and leukocyte (white blood cell) recruitment. Regarding the scavenger function, LSECs possess numerous scavenger receptors that remove blood-borne proteins and lipids, thereby acting as the body’s most powerful scavenger system. Moreover, LSECs release vasodilators, such as nitric oxide (NO), and other molecules that help regulate blood flow in the liver. Additionally, along with bone marrow-derived progenitor cells, LSECs contribute to liver regeneration after injury [8,9].

In disease, LSEC injury in the early stages of NAFLD/MASLD can contribute to the activation of Kupffer cells (KCs) and hepatic stellate cells (HSCs), potentially leading to chronic liver injury, which is the critical step toward fibrosis, cirrhosis, and eventually neoplasm [10,11]. Defenestration (loss of fenestrations) of LSECs occurs during chronic liver disease, contributing to increased portal pressure and the development of liver cirrhosis. Dysregulation of LSECs in cirrhosis can lead to microvascular thrombosis and portal hypertension (cirrhotic portal hypertension). Ultimately, LSECs may play a role in the development and neoplastic progression into hepatocellular carcinoma (HCC), although this aspect has been a topic of controversy to date [9,12,13,14,15,16,17,18,19].

This narrative review examines the distinctive functions of LSECs in regulating immunologic responses, collagenization, and drug-sensitive bioreactivity in a more comprehensive view than before, likely due to our extensive experience of the use of electron microscopy in our laboratories for diagnosis and research. This narrative review scrupulously examines the distinctive functions of LSECs in regulating immunologic responses due to our clinical and experimental immunology experience with liver diseases at our liver transplantation center (University of Alberta, Canada). Moreover, this review highlights that LSECs may be investigated in depth using a human primary 3D model of MASH.

## 2. Liver Sinusoidal Endothelial Cells: Anatomy and Role

The fenestrations of LSECs efficiently filter blood, regulating the entry of lipoproteins, proteins, and immunological components into the liver parenchyma [16,20,21]. The magnitude (dimension) and number of fenestrae vary according to their location in Rappaport’s hepatic lobulus. The periportal LSECs possess larger fenestrations. Moreover, they are fewer in quantity. The pericentral LSECs, on the other hand, have smaller fenestrations. They are more frequently found ultrastructurally [2,22,23]. LSECs, situated between blood flow and hepatocytes, play a crucial role in eliminating infections, toxic products, and waste substances from the bloodstream, including metabolic wastes and conventional cell debris. These waste products encompass chemicals such as bilirubin, ammonia, and various metabolic byproducts that must be effectively eliminated to preserve liver and overall bodily health. LSECs efficiently eliminate circulating antigens, immunological complexes, and apoptotic cells via scavenger receptors, hence enhancing the liver’s function as a detoxifying organ [24]. Figure 1 depicts the architecture and role of LSECs throughout selectin-independent tethering, firm adhesion, intravascular crawling, and transmigration.

LSECs have a crucial role in preserving immunological homeostasis in the liver, in addition to their filtering function. These cells exhibit significant activity in the antigenic cellular presentation, absorbing antigenic molecules from the bloodstream and delivering them to susceptible cells, particularly T lymphocytes, in a manner that generally fosters immunological tolerance rather than immunologic activation [1,12,20]. The immunologic regulatory role of LSECs is essential due to the extraordinary function of the liver as a major body organ. Capillarization is a frequent phenomenon in conventional hepatology. It refers to the transformation of LSECs into cells, which exhibit a phenotype resembling continuous blood capillaries. At the same time, *de*-capillarization signifies the reversal of capillarization, wherein LSECs reclaim their *fenestrae* and almost completely eliminate the cellular basement membrane. The capillarization progression encompasses the elimination of fenestrations and a decrease in tight junctions located between cells. It results in an enhanced barrier integrity. Moreover, the modification in function leads to a greater similarity to microvascular endothelial cells.

Electron microscopic investigations have disclosed structural distinctions between pericentral and periportal LSECs (Figure 2). These investigations also exhibit divergent molecular signatures and activities [26,27,28]. The periportal LSECs primarily facilitate nutritional absorption, cholesterol biosynthesis, and oxidation of lipid molecules, whereas the pericentral LSECs predominantly engage in the metabolism of glycolysis and in xenobiotic metabolism [29].

As indicated above, the liver plays a pivotal role in immunological modulation, serving as a filter to separate blood from the gastrointestinal system, which contains a high concentration of antigenic molecules, nutrients of various origin, and toxins/toxicologic products. In contrast to other organs of the human body, the liver has evolved ways to sustain immunological cell tolerance while concurrently averting infective processes and excessive immunologic reactions. This immunological equilibrium is crucial for thwarting hepatic damage, which could otherwise lead to inflammation (hepatitis) [8,31]. LSECs are anatomically situated adjacent to the sinusoids of the liver cells and are integral to this process [1,2,12].

In contrast to conventional endothelial cells, LSECs exhibit a high degree of specialization in their roles of blood filtration, pathogen clearance, and antigen presentation. They modulate immunological responses by directly interacting with immunologic cells, including T lymphocytes and KCs (also considered resident macrophages), fostering tolerance to benign antigens while facilitating a proper immunologic response to pathogenic threats. LSECs are crucial for maintaining hepatic homeostasis. LSECs act magnificently in preventing immunologic-mediated liver injury [1,2,32,33].

LSECs operate interdependently; their engagement with other liver cells is crucial to liver health. KCs, the liver’s resident macrophages, are among the most crucial cellular partners of LSECs [1,2,6,12,34]. KCs reside in the liver sinusoids and participate in immune surveillance and phagocytosis. LSECs and KCs cooperate to sustain immunological tolerance; for example, LSECs internalize and transmit antigens to KCs, which subsequently assist in mitigating overactive immune responses. KCs react to inflammatory cellular signals from LSECs, coordinating extensive immunologic retorts when required, such as during infective diseases or, generically, liver damage [25,35,36].

LSECs are crucial in the activation, communication, and regulation of HSCs, the principal cells accountable for hepatic fibrosis and cirrhosis. Under typical circumstances, LSECs facilitate the preservation of HSCs in a quiescent state. This setting occurs by secreting soluble substances such as nitric oxide (NO), which inhibits HSC activation [37,38,39]. In the event of liver damage or, specifically, chronic hepatitis (chronic inflammation), LSECs undergo modifications that facilitate HSC activation. These modifications encompass a modified secretion profile, resulting in elevated levels of pro-fibrotic cytokines such as TGF-β, and augmented expression of adhesion molecules such as vascular cell adhesion molecule 1 (VCAM1), facilitating the recruitment of inflammatory cells to the liver. Activated HSCs differentiate into myofibroblasts. Myofibroblasts are able to synthesize collagen and proteins of the extracellular matrix, resulting eventually in fibrosis/cirrhosis. The interaction between LSECs and HSCs is a pivotal element in the advancement of liver disease, particularly in chronic liver disorders such as MASLD [40].

Hepatocytes are the primary cells responsible for liver function, and LSECs also interact with these cells. Nutrient and signaling chemical transport by hepatocytes is dependent on LSECs. Liver cells, or hepatocytes, secrete regulatory factors that help LSECs operate, and fenestrations on LSECs allow important compounds like hormones and lipoproteins to be sent to hepatocytes [9]. In the event of liver injury, the interaction between LSECs and hepatocytes remains crucial, as LSECs react to liver cell damage by triggering immunological responses and regulating fibrosis through their engagement with HSCs [20,21,41]. In diseases such as liver cirrhosis, these cellular interactions become completely dysregulated. They lead to liver malfunction and progressive dysfunction of liver pathophysiology [1,2,33,42].

In fact, LSECs are integral to numerous essential hepatic activities and processes. Their interactions with KCs, HSCs, and hepatocytes govern immune responses and sustain overall liver homeostasis. Comprehending these interactions is essential for discovering novel treatment strategies focused on addressing liver illnesses through the modulation of LSEC-mediated pathways. LSECs are essential for liver regeneration and post-transplant conditions. In the process of liver regeneration, LSECs release growth factors including vascular endothelial growth factor (VEGF) and hepatocyte growth factor (HGF), which facilitate hepatocyte proliferation and tissue repair [43,44]. In liver transplantation, LSECs facilitate immunological tolerance and inhibit graft rejection by promoting regulatory T cells (often labeled Tregs) and decreasing effector T cells. Augmenting the immunomodulatory capabilities of LSECs may enhance organ graft survival and diminish the necessity for prolonged immunosuppression in transplant recipients [41,45,46,47].

Human investigations have shown that the capillarization and de-capillarization of LSECs significantly influence the remission and advancement of liver fibrosis. It has been found that capillarization exacerbates liver fibrosis by compromising the function of LSECs and facilitating the activation of HSCs. On the flip side, improved liver function and regression of fibrosis are associated with de-capillarization and fenestration restoration. A promising treatment strategy for correcting liver fibrosis and improving liver regeneration has been shown to involve drugs which target the capillarization of LSECs [16,48].

## 3. Deep Lens on LSECs’ Immune Modulation

In contrast to conventional endothelial cells, LSECs serve as crucial modulators of hepatic tissue microcirculation and immunologic surveillance. LSECs are essential in regulating the immunologic response by presenting antigens and fostering immunological tolerance within the liver’s distinctive immune milieu, which is perpetually exposed to intestinal-derived antigens and blood-circulating pathogens. This precise equilibrium enables the liver to avert improper immune activation while initiating effective immunological responses when required [6,26,49].

LSECs participate in the presentation of antigenic molecules to immunologic cells, particularly T lymphocytes, a step essential for sustaining immunological tolerance. In contrast to qualified antigen-presenting cells (APCs) like dendritic cells, LSECs possess a distinctive ability to elicit tolerance instead of immunologic activation. They accomplish this by displaying antigens through MHC (major histocompatibility complex) class I and class II molecules to naive T lymphocytes, resulting in the induction of CD8+ T cell tolerance via processes such as clonal deletion and anergy [26,50,51,52,53]. Furthermore, LSECs have diminished expression of costimulatory molecules. They include CD80 and CD86 (CD, cluster of differentiation), which are critical for comprehensive T lymphocytic activation. This lack of costimulation biases T cell responses towards tolerance instead of effector activities. Moreover, LSECs release immunomodulatory substances such as interleukin-10 (IL-10) and TGF-β, which facilitate the maturation of Tregs. In particular, they attenuate pro-inflammatory T lymphocytic responses [1,12,32,54].

Liver illnesses progress along a continuum, beginning with early inflammation and escalating to fibrosis, and then cirrhosis. In the absence of sufficient costimulatory signals, LSECs cross-present foreign antigens to CD8+ T cells, leading to T cell death or anergic condition, which in turn prevents cytotoxic T lymphocyte (CTL) activation and the resulting tissue damage. It is important to emphasize that the preservation of immunological homeostasis in the liver relies on KCs [55,56]. LSECs intimately interact with KCs and other hepatic macrophages to regulate immunological responses. LSECs facilitate the preservation of the liver’s tolerogenic milieu by releasing anti-inflammatory cytokines. KCs, in turn, communicate with LSECs by secreting signaling molecules like IL-10 and TGF-β, hence augmenting the immunosuppressive properties of LSECs [57]. Figure 3 summarizes LSECs’ features during the progression from healthy liver to HCC.

Preventing overactive immunity in the liver relies on the crucial interaction between LSECs and KCs. In some conditions, such as chronic liver disease or infections, KCs can take on a pro-inflammatory phenotype; hence, a disruption of this relationship may lead to increased inflammation (hepatitis) [1,2,6,12,34].

## 4. LSECs’ Contribution to Fibrosis and Cirrhosis

In fibrosis subsequent to steatohepatitis, the extracellular matrix is deposited in the space of Disse, ultimately encasing hepatocytes and augmenting the thickness of this area [32,58]. By fostering tolerance, LSECs avert superfluous immunological responses that may harm the liver, rendering them essential for hepatic immune monitoring and the preservation of systemic immune equilibrium. The incidence of MASLD/MASH is escalating worldwide, particularly in the United States and Canada [59,60,61]. By 2030, it is projected that approximately 27 million adults in the US will have MAFLD. This increase is directly associated with the escalating prevalence of obesity and T2DM, which are significant risk factors for the onset of MASLD [62,63,64]. Clinically, MASH is difficult to detect because of the absence of diagnostic markers for early liver fibrosis, resulting in frequent underdiagnosis. The advancement of fatty liver can lead to considerable hepatic damage, requiring prompt identification and care to avert serious consequences. Comprehending the prevalence rate, risk factor ratios, and clinical importance of MASLD remains essential for formulating effective public health measures and therapeutic interventions to address and alleviate this escalating health issue [27,65]. MASH represents the more severe variant of MASLD, encompassing inflammation and damage to liver cells, including LSECs, which may progress to fibrosis, cirrhosis, and HCC [5,66].

In investigating the immunologic interactions of LSECs, we found that TGF-β exhibits a two-fold function in liver pathophysiology, serving as both a pro-fibrotic and anti-inflammatory cytokine. In typical LSECs, TGF-β expression facilitates immunological homeostasis. During chronic liver damage, the overexpression of activated LSECs stimulates HSCs activation and extracellular matrix deposition, hence facilitating the advancement of fibrosis. The immune-regulatory role of LSECs is essential due to the liver’s continuous exposure to food and bacterial antigens from the gastrointestinal tract [67,68,69,70]. By fostering tolerance, LSECs avert superfluous immunological reactions that may harm the liver, rendering them essential for hepatic immune monitoring and the preservation of systemic immune equilibrium, and abnormalities of responses of tumor suppressor genes.

The buildup of extracellular matrix proteins, a defining feature of chronic liver illnesses and a precursor to more serious outcomes including cirrhosis and liver failure, is a characteristic of liver fibrosis, which develops in advanced MASLD. Liver function is impaired in fibrosis due to the progressive replacement of normal liver architecture by scar tissue [71]. This process is predominantly facilitated by HSCs, which are stimulated in reaction to hepatic damage. LSECs are crucial to this process through their interactions with HSCs and immune cells. In a healthy liver, LSECs preserve the quiescent state of HSCs by releasing regulatory chemicals. During liver injury or chronic hepatitis (chronic inflammation), LSECs experience phenotypic alterations that facilitate the activation of HSCs, resulting in fibrosis [1,2,12]. As fibrosis advances, it undermines the liver’s regeneration ability and may eventually lead to end-stage liver disease (ESLD). Comprehending the roles of LSECs in immune modulation and fibrosis is essential for formulating therapeutic approaches aimed at mitigating liver damage and enhancing patient outcomes [1,2,12].

In liver fibrosis, LSECs exhibit behavior akin to conventional vascular endothelial cells, diminishing their capacity to eliminate antigens and uphold immunological tolerance. The impairment of LSECs fosters a pro-inflammatory milieu in the liver, resulting in increased influx of immune cells such as monocytes and neutrophils. Chronic inflammation progressively results in the excessive accumulation of extracellular matrix by activated HSCs, promoting fibrosis and ultimately cirrhosis [1,2,6,12]. Antigen presentation, contact with immune cells, and the release of anti-inflammatory cytokines are three ways in which LSECs keep the liver’s immunological tolerance intact.

Nitric oxide (NO) aids in maintaining hematopoietic stem cells in an inactive state and inhibits the overproduction of extracellular matrix proteins, such as collagen. In the context of chronic liver injury, whether resulting from alcohol addiction or MASLD, LSECs experience functional alterations that activate hematopoietic stem cells [2,33,72]. When LSECs become dysfunctional, they forfeit their capacity to generate adequate NO and other antifibrotic signals. Consequently, HSCs are activated and differentiate into myofibroblasts, which are accountable for the secretion of substantial quantities of collagen and other extracellular matrix components. The excessive accumulation of collagen disturbs the liver’s normal structure, resulting in scarring and facilitating the progression to cirrhosis. Furthermore, the capillarization of LSECs intensifies fibrosis by limiting the delivery of nutrients and oxygen to hepatocytes, thereby exacerbating liver damage and chronic inflammation (hepatitis) [73]. Of note, LSECs become increasingly vulnerable to TGF-β, leading to their capillarization and dysfunction [1]. A distinguished pathway is the VEGF signaling system, which has been studied in pediatric laboratories in connection with eosinophilic esophagitis as well as pediatric cancer [74,75,76]. Under typical circumstances, VEGF preserves the fenestrated cellular architecture of LSECs and promotes their viability. In chronic liver illness, diminished VEGF signaling results in the capillarization of LSECs, which contributes to fibrosis [1,2,12].

Oxidative stress is a crucial element in LSEC-induced fibrosis [77,78]. In circumstances like alcoholic fatty liver disease or non-alcoholic fatty liver disease, heightened oxidative stress results in the generation of reactive oxygen species (ROS) molecules. Hydrogen peroxide, water, and molecular oxygen (O_2_) are the building blocks of reactive oxygen species (ROS). Hydroperoxide (H_2_O_2_), superoxide (O_2_⁛), and hydroxyl radicals (•OH) are important members of this group.

ROS molecules impair LSECs and contribute to their dysfunction. This further activates pro-fibrotic signaling pathways, including platelet-derived growth factor (PDGF), which further promotes HSC activation and progressive collagen synthesis (fibrosis–cirrhosis) [77,78,79].

## 5. LSECs’ Contribution to “Pro-Carcinogenetic” Cirrhosis

Research indicates that LSECs from fibrotic livers demonstrate elevated levels of TGF-β and reduced expression of NO synthases. The alterations have been associated with increased HSC activation and collagen accumulation, demonstrating a direct involvement of LSEC malfunction in the evolution of fibrosis [1,2,54]. Restoring NO levels in LSECs mitigated fibrosis by decreasing HSC activation. Likewise, several investigations highlighted that LSECs that underwent capillarization exhibited diminished efficacy in blood filtration and immune cell recruitment regulation. The absence of fenestrations in LSECs resulted in increased recruitment of inflammatory macrophages, which subsequently activated HSCs and, eventually, exacerbated liver fibrosis [1,2,33,42,54]. As MAFLD advances to MASH, LSECs undergo capillarization and acquire traits similar to vascular endothelial cells. Because of their failure to control immune responses, scavenging activities, and portal pressure, LSECs consequently exacerbate inflammation and fibrosis. Inflammatory cytokines including Tumor Necrosis Factor-alpha (TNF-α) and interleukin 6 (IL-6) have been found in malfunctioning LSECs, exacerbating liver damage and local inflammation [80,81,82]. Increased collagen deposition and fibrosis are outcomes of these inflammatory signals, which further activate HSCs. From the early stages of inflammation to the more advanced stages of fibrosis and cirrhosis, LSECs interact dynamically with other hepatic cells. In order to affect the outcomes of liver injury and the rate of advancement of chronic liver diseases, this cellular communication cannot be altered.

However, in the initial phases of MAFLD, LSECs continue to serve as a protective function by upholding immunological tolerance and inhibiting excessive immune activation. LSECs extract antigens from the circulatory system and interact with KCs to regulate immunological responses. Nevertheless, as liver damage continues, LSECs forfeit their protective function. Inflammatory signals from injured hepatocytes and invading immune cells, including macrophages and neutrophils, stimulate LSECs to secrete pro-inflammatory and pro-fibrotic substances, which have also a pro-carcinogenetic effect [1,2,12].

As inflammation persists, LSECs increasingly engage with HSCs in a more pathogenic manner. During liver damage, LSECs diminish NO generation while augmenting the secretion of TGF-β and PDGF. These signals stimulate HSCs, prompting their differentiation into myofibroblasts, which synthesize collagen and other extracellular matrix components that contribute to fibrosis/cirrhosis. LSECs facilitate further fibrosis via promoting tissue angiogenesis, which is regulated by VEGF. The atypical development of blood vessels induces a hypoxic environment that sustains liver fibrosis and impairment. At this juncture, the interaction among LSECs, HSCs, and immune cells evolves inevitably into a detrimental cycle characterized by persistent collagen tissue deposition, inflammatory changes, and hepatic tissue remodeling. Comprehending the mechanisms underlying LSEC failure and their interactions with other hepatic cells presents prospective therapeutic targets to impede or reverse the progression of liver disease. Cirrhosis may, in certain instances, advance to HCC [2,48,83,84,85,86]. LSECs play a complex and multifaceted role in the development and progression into HCC. While healthy LSECs contribute to maintaining liver homeostasis and immune tolerance, their dysfunction can promote inflammation, fibrosis, and ultimately, cancer.

LSECs can both inhibit and promote HCC depending on the stage of the disease and the specific interactions within the tumor microenvironment. During the early stages of HCC development, LSECs may still contribute to tumor suppression by interacting with immune cells, such as CD8+ T cells, and promoting immune tolerance. However, in later stages, LSECs can become involved in creating a dreadful tumor-promoting microenvironment. They can interact with tumor cells, promote angiogenesis, and contribute to the formation of an immunosuppressive environment that allows tumors to grow and spread. The capillarization of LSECs is closely linked to HCC development. It can lead to increased intrahepatic vascular resistance, portal hypertension, and a pro-inflammatory environment that ineluctably favors tumor progression. Overall, LSECs seem to be the culprit dynamic cells that play a crucial role in the development and progression of HCC. While they can contribute to tumor suppression in the early stages, their dysfunction and interaction with tumor cells can promote HCC development, angiogenesis, and metastasis. Thus, targeting LSECs holds promise for developing new therapies for HCC, particularly in the context of chronic liver diseases. In fact, therapeutic efforts may concentrate on targeting tumor vasculature and reducing angiogenesis, potentially employing anti-angiogenic drugs and immune modulators in the future using nanotechnology [21].

## 6. LSECs’ Potential Treatment Approaches

One promising new way to treat liver fibrosis and immune system abnormalities is by targeting LSECs. This section examines prospective therapeutic strategies directed at LSECs and evaluates current and novel treatments designed to improve LSECs’ functionality or impede the fibrotic cascade [21]. In optimal settings, LSECs display fenestrations that enable the transfer of substrates between the bloodstream and hepatocytes. Therapeutic approaches designed to restore the fenestrated phenotypic and functional integrity of LSECs are therefore crucial for critical investigations. Agents like VEGF and NO can sustain the fenestrations of LSECs and inhibit capillarization. Augmenting VEGF signaling has been demonstrated to preserve the differentiation and function of LSECs [43,87,88]. LSECs react to shear stress generated by blood circulation. Altering shear stress via mechanical or pharmacological methods can affect the phenotype of LSECs and inhibit the advancement of fibrosis. Targeting the signaling pathways implicated in LSEC-mediated HSC activation is a feasible therapeutic strategy. TGF-β is a crucial cytokine implicated in the stimulation of hematopoietic stem cells. Inhibiting TGF-β signaling in LSECs can diminish their pro-fibrotic impact on hematopoietic stem cells. TGF-β plays a dual role in immune function and liver disease. As an anti-inflammatory cytokine, it facilitates immunological tolerance by enhancing regulatory T lymphocyte development and inhibiting pro-inflammatory responses. This process is essential for preserving immunological homeostasis in normal LSECs and mitigating inflammation-induced tissue damage during the initial phases of liver disease [41,45,47]. In addition, the Notch signaling pathway in LSECs affects vascular remodeling and fibrogenesis. Modulating Notch signaling may reduce fibrotic responses [89,90,91].

In order to maintain tolerance and immunological surveillance in the liver, LSECs are crucial. A dysregulated immune system is worsened by LSEC failure in liver fibrosis. Restoring proper immunological control by LSECs is one potential therapeutic strategy for fibrosis and inflammation alleviation. Modulation of immunological checkpoints may be essential [31,92,93,94].

The administration of anti-inflammatory molecules or inhibitors of pro-inflammatory cytokines can restore the immunological environment to facilitate fibrosis resolution. Numerous pharmaceuticals are now being studied or used to enhance the functionality of LSECs, with the objective of preventing or reversing liver fibrosis. In addition to their lipid-lowering properties, statins have demonstrated the ability to enhance endothelial function. In LSECs, statins can augment NO synthesis, preserve fenestrations, and impede HSC activation. Clinical investigations indicate that statin medication may impede the progression of fibrosis in cirrhosis. Although angiogenesis is frequently linked to pathological states, regulated restrictions can avert abnormal vascular remodeling in fibrosis. Agents that target VEGF receptors may assist in preserving the formation and function of LSECs [43,95,96]. Agonists of the Farnesoid X receptor (FXR), including Ocaliva^TM^ or obeticholic acid, exhibit hepatoprotective and antifibrotic properties. FXR agonists regulate bile acid metabolism and demonstrate anti-inflammatory characteristics that indirectly enhance the action of LSECs [97,98]. Obeticholic acid, a drug for liver illness made by Intercept Pharmaceuticals, is now under investigation by the FDA since it has been found to cause liver damage in people who do not yet have severe scarring.

The fibrotic cascade is the primary focus of new therapeutic treatments, with LSECs playing a key role in these strategies. By blocking fibrogenic pathways, including those promoted by LSECs, the approved drugs pirfenidone and nintedanib show promise in the treatment of liver fibrosis. These drugs are mostly used to treat idiopathic pulmonary fibrosis. Fibrogenesis and inflammation are processes in which galectin-3 is involved. Reducing HSC activation and increased production of extracellular matrix by inhibiting galectin-3 improves LSEC function. To enhance therapy outcomes, more potent and specific inhibitors are now in developed. By differentiating into functional LSECs, mesenchymal stem cells (MSCs) and endothelial progenitor cells (EPCs) can promote vascular regeneration and reduce fibrosis.

Gene therapy involves the administration of genes that encode protective factors, such as VEGF or antifibrotic proteins, to LSECs to augment their regeneration potential and suppress fibrogenic signaling [65]. Nanocarriers can be engineered to target LSECs for the precise delivery of therapeutic drugs, thereby improving treatment efficacy and reducing off-target effects. LSEC-targeted nanoparticles using ligands that specifically bind to receptors uniquely expressed on LSECs, such as mannose receptors, facilitate the targeted delivery of antifibrotic medicines or siRNA molecules to these cells [21]. Nanotechnology-enabled devices facilitate the sustained release of therapeutic drugs, providing extended regulation of LSECs and suppression of fibrosis. The identification of innovative macromolecules and small compounds that precisely regulate the function of LSECs is an emerging area of research. MicroRNA modulators (miRNAs) govern gene expression in LSECs. Therapeutics that replicate or obstruct certain miRNAs can modify the function of LSECs to promote antifibrotic results. Identifying small compounds that impede pro-fibrotic enzymes or signaling molecules will be critical in the future.

Finally, a study on the development of biomarkers examines the application of an automated artificial intelligence (AI) platform or automated machine learning (AutoML) diagnostic support system, such as a computational biomarker for identifying drug-induced liver damage (DILI) patterns in hepatopathology, and the liquid biopsy approach is promising. The new assay effectively classifies necrotic damage patterns with high precision, serving as a valuable tool for the early detection and assessment of hepatotoxicity in drug development, and nucleic acid spheres for treating capillarization of LSECs in liver fibrosis is highly promising [54].

Table 1 and Table 2 show several approaches to different degrees of liver damage as well as prospective directives.

## 7. Three-Dimensional Human Model

MASLD encompasses a range of conditions defined by an accumulation of fat in the liver (steatosis), affecting roughly 30% of the global population. Soon, MASH is projected to exceed hepatitis C as the primary reason for liver transplants in the United States. The mortality rate for patients with MASH is 7.9%, which is double that of the general population. The prevalence of MASH in the United States is projected to double every decade, with an estimated 43 million Americans anticipated to be affected by the disease by 2025. In several studies, LSEC isolation has been critical for most of our current knowledge. It is evident that LSECs need to be isolated properly for critical investigations. LSECs can be isolated using several methods, including elutriation, Percoll gradient separation with selective adherence, and immunomagnetic separation. These methods aim to separate LSECs from other liver cell types like hepatocytes and KCs, enabling further study of their unique functions. Elutriation consists of separating cells based on size and density using a centrifugal force and a countercurrent flow of buffer. It can yield high numbers of LSECs with high purity, but the process is technically demanding and requires specific equipment. Percoll gradient separation with selective adherence is a method which involves separating cells based on their density using a Percoll gradient, followed by allowing LSECs to selectively adhere to a culture dish or other surface. It is a widely used method but yields and purities can vary. Immunomagnetic separation uses antibodies attached to magnetic beads to bind to specific surface markers on LSECs, allowing for their separation using a magnetic field. It is a relatively fast and efficient method but can sometimes result in lower cell yields and potentially identify subpopulations of LSECs due to the use of specific surface markers, according to the National Institutes of Health (NIH). Some protocols avoid perfusion and instead use chopped liver tissue, which can be beneficial in certain situations like studying LSECs from fibrotic livers or higher vertebrate species. Advances in cell sorting techniques (e.g., magnetic and fluorescence-activated cell sorting) have enabled the development of high-throughput methods for isolating LSECs. Researchers have also developed methods for cryopreserving LSECs, which allows for long-term storage and the ability to use them later.

In vivo models using rodents to study human LSECs have obvious limitations due to significant species-specific differences in liver biology and function, as explained below. These differences impact the ability to accurately translate findings from rodent models to humans, particularly in areas like immune responses, drug metabolism, and susceptibility to certain pathogens. Key limitations include (1) interspecies differences in liver biology (human and rodent livers exhibit structural and functional variations, including differences in liver zonation, cytokine responses, and the specific types and proportions of immune cells), (2) immune cell differences (the human liver has a greater abundance of NK cells compared to mice, and certain immune cell interactions, like those involving IL-8, are not conserved between species), (3) drug metabolism and toxicity (rodent models may not accurately reflect human drug metabolism pathways, leading to inaccurate predictions of drug efficacy and toxicity in humans), (4) pathogen susceptibility (rodents may not be susceptible to the same pathogens that cause human liver diseases, limiting the utility of rodent models for studying infectious liver diseases), (5) limited repopulation and function of human hepatocytes in humanized mice (while humanized mice, where human liver cells are transplanted into immunodeficient mice, are used to study human liver diseases, they often exhibit limitations in the repopulation rate of human hepatocytes and the full functionality of human immune cells, (6) challenges in culturing and maintaining LSECs (both human and rodent LSECs are challenging to culture and maintain in vitro, with human LSECs having limited passaging capacity and rodent LSECs dying shortly after isolation, (7) incomplete conservation of liver-specific lectins (some liver-specific lectins, crucial for cell interactions and immune responses, are not conserved between mice and humans, affecting the accuracy of studying these interactions in rodent models), and (8) variations in bile acid metabolism (humans and mice have different bile acid profiles, which can affect the activation of nuclear receptors like FXR, impacting bile acid-related liver functions. Moreover, 2D cell cultures, including those of hepatocytes and LSECs, lack the complexity of the liver’s multicellular environment, leading to limitations in accurately modeling liver function and drug responses.

Thus, a 3D human model could be highly beneficial. Three-dimensional human liver models aim to replicate several studies in vitro. These models often involve co-culturing LSECs with other liver cells like hepatocytes and hepatic stellate cells, mimicking the complex cellular interactions found in the liver. This approach helps researchers study liver physiology, drug responses, and disease mechanisms more accurately than traditional 2D models. Three-dimensional models, unlike two-dimensional cultures, can better represent the layered structure of hepatic sinusoids and the interactions between different liver cell types. Co-culturing LSECs with hepatocytes and other cells (like hepatic stellate cells) allows for the study of cell-to-cell communication and its impact on liver function. Three-dimensional models may incorporate scaffolds or matrices (e.g., collagen) to provide more in vivo-like environments and support cell growth and function. Various 3D culture paradigms exist, including micropatterned co-cultures, microcarrier bead configurations, matrix-embedded cultures, and bioprinted models. Co-culturing LSECs with hepatocytes in 3D can enhance hepatocyte function and maintain their phenotype for longer periods compared to 2D cultures. Three-dimensional models, including those with LSECs, can provide more accurate predictions of drug-induced liver toxicity and efficacy. These models can be used to study various liver diseases, such as liver fibrosis and drug-induced liver injury. Three-dimensional models can be developed using cells from individual patients, potentially paving the way for personalized treatment strategies. Researchers are developing strategies to generate LSECs from human pluripotent stem cells (hPSCs), providing a renewable source of cells for research and potentially therapeutic applications. Techniques include optimizing cell differentiation protocols to promote LSEC fate from angioblasts (cells that develop into blood vessels). These hPSC-derived LSECs can be incorporated into 3D liver models to study liver biology and disease. In essence, 3D human liver models incorporating LSECs offer a powerful tool for advancing our understanding of liver biology, disease mechanisms, and drug response. They provide a more physiologically relevant environment for studying liver cells and their interactions, leading to more accurate preclinical models for drug development and disease research.

Three-dimensional bioprinted liver models utilizing primary human liver cells have been created that demonstrate tissue-like density and well-organized cellular characteristics, such as intercellular tight junctions, microvascular networks, and a microenvironment conducive to “in vivo”-like cellular function, surpassing the capabilities of two-dimensional monocultures or monolayer co-cultures, while also maintaining a more defined architecture compared to self-aggregated co-culture models. Three-dimensional bioprinted tissues preserve metabolic functions in culture, including the activities of key enzymes and molecular transporters, without significant decline for several weeks. The reactions of 3D bioprinted tissues to acute or chronic exposure to pharmaceuticals and established toxins are analogous to in vivo liver tissue. The automated bioprinting process produces scalable tissues, ensuring precise control over composition and morphology. Prior 3D bioprinted models utilized cells from healthy donors, specifically human umbilical vein endothelial cells (HUVECs) instead of LSECs, along with additives like transforming growth factor-β to simulate the disease process.

Atomic Force Microscopy (AFM) assesses elasticity by acting as a nanoindenter, allowing the measurement of the Young’s Modulus (or elastic modulus) of materials from individual cells to tissues and polymers. Multiple methodologies have been proposed for utilizing AFM to assess elasticity, cytoskeletal organization, cellular dimensions, and height, as well as the arrangement of membrane proteins [99,100,101,102,103,104]. The range of departures from the physiological circumstances of endothelial cells has been demonstrated to be somewhat indicative of the biophysical features of the examined cells. For instance, alterations in both the shape and nanomechanical characteristics of endothelial cells have been meticulously recorded during inflammation, hyperglycemia, or hypertension [105]. A recent review detailed the advancements achieved thus far regarding the different types of LSECs and hepatocytes, utilizing AFM. A succinct assessment of the AFM–liver literature reveals that this probing approach facilitates the monitoring of the biophysical properties of liver cells under various experimental settings with unparalleled resolution across multiple magnitudes (X, Y, Z, and t, hereafter referred to as 4-D). Utilizing a 4-D AFM methodology, some authors applied AFM to assess the progression of experimentally induced fatty liver disease in murine models [101]. Size and diameter variations can serve as morphological indications of health state. Nanometer-scale transcellular pores have been observed to respond to various external stimuli, serving as a significant indicator of the functional responsiveness of LSECs. The nanomechanical characteristics of this fenestrated sinusoidal endothelium under diverse experimental settings during live cell imaging was also recorded [105]. Moreover, substantial advancements were achieved in the comprehensive examination of the cytoskeletal organization related to fenestrae and sieve plates. In 2017, Zapotoczny et al. demonstrated that recent advancements in AFM may be effectively utilized for imaging both fixed and live LSECs [102,103,106,107]. The methodology, which utilizes the collection of force–distance curves, facilitated the minimization of lateral forces, thereby permitting thorough examinations of delicate samples. For a given loading force, the cell surface topography may be recreated from the contact point all the way to the maximum loading force used for the measurement.

Choosing high spatial resolution for extensive picture acquisition led to a diminished imaging speed, which was slower than the dynamics of cellular natural processes, resulting in distorted photographs. Consequently, when high-resolution imaging (i.e., point-to-point resolution <40 nm) was required, an area of several μm^2^ was generally chosen. Enhanced contrast revealing surface details can be achieved using a reconstruction of cell topography rather than through stiffness calculations. Utilizing a mild loading force of around 200–300 pN, pictures of cell topography distinctly reveal fenestrae.

A reduced loading force facilitated the minimization of topographical abnormalities in the studied cells, resulting from deformation induced by the pressure applied by the tip on the (sub)membranous cytoskeletal structures. Various results indicate that the indentation caused by a loading force of 200–300 pN corresponds to the cell’s cortical layer. Increased loading forces (>500 pN) integrate the mechanical response of the cortex and the deeper cellular components, including the nuclear envelope. Forces applied to LSECs within the allowed range cause actin stress fibers and fenestrae-associated cytoskeleton rings (FACRs) to develop.

Recently, innovative optical nanoscopy techniques, such as stimulated emission depletion microscopy (STED), three-dimensional structured illumination microscopy (3DSIM), and direct stochastic optical reconstruction microscopy (dSTORM) have successfully resolved *fenestrae* in isolated LSECs.

## 8. Present Obstacles and Prospective Pathways

Despite considerable progress in comprehending the function of LSECs in liver disease, substantial gaps remain that impede the formulation of effective therapies aimed at these cells. A principal obstacle resides in the intricacy of investigating LSECs within the framework of liver illness, primarily attributable to the constraints of in vivo models. For example, rodent LSECs exhibit significant physiological differences from human LSECs, especially for their immune-regulatory roles, complicating the realistic modeling of immunological tolerance and disease development. Furthermore, most current models fail to accurately simulate chronic liver diseases, such as cirrhosis, in which prolonged interactions between LSECs and other hepatic cells are essential for disease advancement. Moreover, clinical investigations examining the particular functions of LSECs in liver disease remain scarce. The majority of research on LSECs has occurred in preclinical environments, and the quantity of human trials targeting LSECs therapeutically is still limited. Although LSECs play a crucial role in liver fibrosis and immunological modulation, clinical research investigating their direct participation in human liver disorders remains insufficient. The variability of LSECs across various stages of liver disease, particularly during the progression from a healthy to a pathological condition, remains inadequately comprehended. Additional clinical research is required to clarify how alterations in the phenotype and function of LSECs influence illness outcomes and to find possible biomarkers that may forecast fibrosis development or liver dysfunction. Comprehending the mechanics of LSEC failure offers a detailed foundation for creating tailored therapeutics to alleviate liver disease development. Novel research domains, like organoids and computer modeling, present exciting opportunities for investigating the function of LSECs. Organoids, three-dimensional cell culture systems that replicate the structure and function of organs, and assembloids, 3D cellular structures created by combining multiple organoids, spheroids, or other cell types to mimic the structure and function of organs, can yield significant insights into the behavior of LSECs in a controlled setting. Conversely, computational modeling facilitates the simulation of intricate biological processes, permitting researchers to anticipate the consequences of LSEC malfunction and pinpoint prospective therapy targets. These unique methodologies possess the capacity to enhance our comprehension of LSEC-mediated disease and guide the formulation of novel therapies.

There is an increasing necessity to comprehend the potential functions of LSECs in liver regeneration and transplantation. LSECs demonstrate potential in facilitating liver regeneration through their capacity to sustain immunological tolerance and regulate inflammation. Studies indicate that LSECs facilitate hepatocyte proliferation through the secretion of growth factors. The processes by which LSECs facilitate liver regeneration, particularly in a chronically injured liver, are not well comprehended. Further research is necessary to clarify these mechanisms and investigate how LSECs might be modified to improve liver regeneration after injury or surgical resection. Examining the regeneration capacity of LSECs may yield innovative approaches for treating liver illnesses that presently necessitate liver transplantation.

## 9. Conclusions

In summary, LSECs are crucial in controlling liver function, especially regarding immunological responses and fibrosis in chronic liver disorders. This review emphasizes the distinctive structure and exquisitely specific functions of LSECs, enabling them to serve as essential regulators of liver tissue homeostasis. In addition to filtering blood and eliminating pathogens, LSECs deliver antigens and promote immune tolerance, which they use to help govern the immune system. Ensuring immunological balance and avoiding excessive inflammation relies on their interaction with other hepatic cells, such as hepatocytes, HSCs, and KCs. But when LSECs are damaged, liver diseases progress more easily. When HSCs activate and collagen accumulates due to dysfunctional LSECs, liver scars and fibrosis develop, and the illness advances. This review’s most important finding is that LSECs actively regulate the immunological milieu and fibrotic response in the liver, rather than just being passive endothelial cells. Dysregulated immune responses and fibrogenesis are the outcomes of LSEC phenotypic changes. Accordingly, LSECs are crucial mediators and potential therapeutic targets in other chronic liver diseases. There is a need to include LSECs in liver pathology due to their ability to control immune responses, interact with HSCs, and influence the course of fibrosis. So, LSECs are a potential therapeutic target for regulating the immune system and preventing fibrosis. The restoration of normal LSEC function and inhibition of fibrosis might be achieved by focusing on processes related to their dysfunction, including capillarization, NO production, and TGF-β signaling.

Moreover, augmenting the regeneration capacity of LSECs or leveraging their immune-regulatory functions may offer novel approaches for managing chronic liver disorders and boost liver transplantation outcomes. As research progresses in elucidating the intricate functions of LSECs, their potential for therapeutic applications is expected to significantly influence the development of future treatments for liver illnesses.

## Figures and Tables

**Figure 1 ijms-26-08006-f001:**
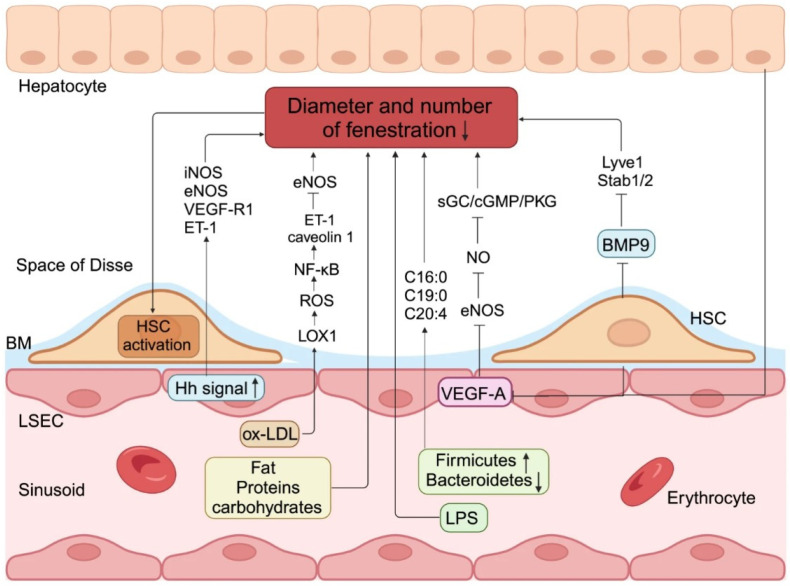
The pathogenic process behind capillarization in liver sinusoidal endothelial cells (LSECs). Excessive consumption of dietary fats, proteins, or carbs, together with alterations in intestinal bacteria or lipopolysaccharides (LPS), can lead to a reduction in both the diameter and quantity of liver sinusoidal endothelial cell (LSEC) fenestrations. Moreover, exposure to ox-LDL elevated the expression of lectin-like ox-LDL receptor 1 at both the mRNA and protein levels, enhanced ROS production and NF-κB activation, subsequently upregulated ET-1 and caveolin 1 expression, and downregulated eNOS expression, thereby diminishing fenestra diameter and porosity. Aberrant activation of Hh signaling in LSECs correlates with elevated production of iNOS, eNOS, VEGF-R1, and ET-1, alongside LSECs capillarization in vitro. Similarly, when the disturbance of the local endocrine milieu results in a reduction in VEGF-A production by hepatocytes and HSCs, nitric oxide downstream of eNOS causes a substantial closure of the fenestrations in LSECs via the sGC/cGMP/PKG pathway. Finally, diminished release of BMP9 produced by HSCs markedly elevates the quantity of basal layer deposition and decreases the amount of fenestrae by downregulating terminal differentiation markers of LSECs. The capillarization of LSECs results in the loss of gatekeeper function, subsequently activating HSCs [Source: He, Q., He, W., Dong, H. et al. Role of liver sinusoidal endothelial cell in metabolic dysfunction-associated fatty liver disease. *Cell Commun Signal* **22**, 346 (2024). https://doi.org/10.1186/s12964-024-01720-9] [25].

**Figure 2 ijms-26-08006-f002:**
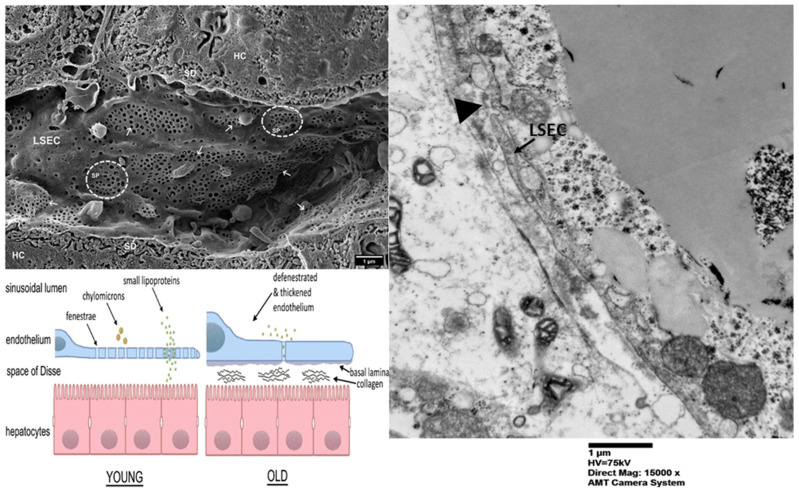
Composite figure: the top left side presents a scanning electron microscopy (SEM) image of hepatic sinusoids in a C57BL6 mouse, approximately 4 months old. Liver sinusoidal endothelial cells (LSECs) exhibiting multiple fenestrations (indicated by arrows) organized into sieve plates (SPs, delineated by dotted-line circles) throughout the sinusoids. SD denotes the space of Disse; HC refers to hepatocytes. The sinusoidal lumen undergoes changes with age, as the fenestrated architecture is compromised during the process of “pseudocapillarization.” Furthermore, the endothelium becomes thicker, and collagen deposits are observed within the space of Disse, resulting in impaired transfer between blood and hepatocytes (source: Szafranska K, Kruse LD, Holte CF, McCourt P and Zapotoczny B (2021) The wHole Story About Fenestrations in LSEC. *Front. Physiol.* 12:735573. doi: 10.3389/fphys.2021.735573) [30]. The right side of the composite figure shows a transmission electron microscopic (TEM) examination of the liver of a pediatric patient, who went for liver biopsy and was found to have a “CRN score” of 7, displaying some obliteration of fenestrae (scale bar: 1 micrometer, direct magnification 15,000×). The CRN score refers to a scoring system developed by the NASH Clinical Research Network for assessing the severity of liver damage in patients with non-alcoholic fatty liver disease. The electron microscopic photograph belongs to the personal archive of the senior author (CMS). The photograph is fully anonymized. No copyright issues are present.

**Figure 3 ijms-26-08006-f003:**
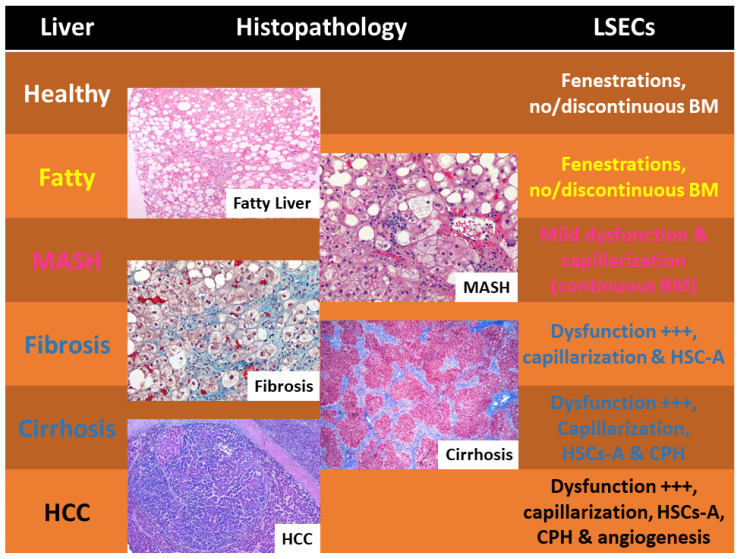
LSECs and liver histopathology. In healthy livers, fenestration with no or discontinuous basement membrane (BM) is a common finding, but dysfunction of these cells, capillarization, and activation of the hepatic stellate cells (HSC-A/HSCs-A) is common in fibrosis following fatty liver and MASH. Cirrhosis discloses portal hypertension, while neo-angiogenesis characterizes the development of abnormal vascular architecture of neoplastic clones in the liver (HCC, hepatocellular carcinoma). All microphotographs depicted in Figure 3 are derived from the personal archive of the senior author (CMS). CPH, cirrhotic portal hypertension. No copyright issues are present.

**Table 1 ijms-26-08006-t001:** LSEC drugs and further strategies.

Stage of Liver Disease	Therapeutic Interventions/Drugs
**Healthy Liver**	Healthy lifestyle, toxin-free nutrition, antioxidants, exercise
**Fatty Liver**	Lifestyle intervention, diet, daily exercise
**MASH**	Lifestyle intervention, diet, daily exercise, resmetirom, GLP-1 receptor agonists, and PPAR agonists
**Fibrosis**	ITDs [e.g., ASK1 or NF-κB ITDs (cenicriviroc, a dual CCR2/CCR5 inhibitor)], ECM Synthesis and Degradation drugs [BMS-986263, an HSP47 mRNA inhibitor, and agents that target galectin-3 (e.g., GR-MD-02)], liver cell-protecting drugs (e.g., emricasan, a pan-caspase inhibitor), REDs (e.g., liraglutide and semaglutide, GLP-1 analogs used for T2DM), others (pioglitazone, a PPARγ agonist, obeticholic acid, a FXR agonist, ARBs (e.g., losartan and candesartan), silymarin)
**Cirrhosis**	NSBBs (e.g., propranolol, nadolol, carvedilol), diuretics (e.g., spironolactone, furosemide), laxatives (lactulose or rifaximin), SBP antibiotics, MTDs (e.g., rifaximin), LTAIs, and statins
**HCC**	Anti-angiogenesis factors (sorafenib and ramucirumab) and immunomodulators [ICIs (e.g., atezolizumab (anti-PD-L1), bevacizumab (anti-VEGF), nivolumab (anti-PD-1), pembrolizumab (anti-PD-1), ipilimumab (anti-CTLA-4)), oncolytic viro-immunotherapy, adoptive T-cell transfer, and vaccines]

Notes: ARBs, angiotensin II receptor blockers; CPH, cirrhotic portal hypertension; ICIs, Immune Checkpoint Inhibitors; FXR, Farnesoid X receptor; ITDs, inflammation-targeting drugs; ASK1, apoptosis signal-regulating kinase 1; REDs, Repurposing Existing Drugs; non-selective beta-blockers (NSBBs), spontaneous bacterial peritonitis (SBP); LTAIs, long-term albumin infusions; MTDs, microbiome-targeting drugs.

**Table 2 ijms-26-08006-t002:** LSEC therapeutic approaches and directives.

Therapeutic Approaches	Technology-Related Drugs	Directives
Vascular Modulation	Angiogenesis inhibitors, statins, FXR agonists	LB-based biomarker development (LBBD)
TGF-β Signaling	TGF-β inhibitors	RCTs on LSEC pathophysiology
Notch Pathway	Controlled release-associated nanoparticles, Notch inhibitors	RCTs on LSEC pathophysiology
Shear Stress	SSMs (Pirfenidone, Nintedanib, Galectin-3 inhibitors)	RCTs
Immune Checkpoint	ICMs	LSEC-microRNA modulators
Cytokine Interplay	Cytokine inhibitors	LSEC-directed nanoparticles

Notes: LB, liquid biopsy; LBBD, identification of reliable biomarkers for LSEC dysfunction using liquid biopsy, patient stratification, and therapeutic responses monitoring; SSM, shear stress modulators.

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
