# Peer review of "Liver Sinusoidal Endothelial Cells and Their Regulation of Immunology, Collagenization, and Bioreactivity in Fatty Liver: A Narrative Review"

_ijms, 2025, doi:10.3390/ijms26168006_

Round 1
Reviewer 1 Report
Comments and Suggestions for Authors
In this review, the authors summarize the progress on the effects of liver sinusoidal endothelial cells in regulating immunologic responses, collagenization, and drug-sensitive bioreactivity in healthy, metabolic dysfunction-associated steatotic liver disease, metabolic associated steato-hepatitis, as well as a human primary 3D model. The content of this review is useful to guide liver sinusoidal endothelial cells-related researches.
This manuscript needs some modifications:
- The tile of this review is too long, which should be refined.
- Please carefully check the writing of the full names and their abbreviations, especially the capital letter and lowercaseletter. For example, Liver Sinusoidal Endothelial Cells (LSECs)should be written as “Liver sinusoidal endothelial cells (LSECs)” in the abstract.
- Abstract is confusing and not well organized, especially the impacts of liver sinusoidal endothelial cells in regulating immunologic responses, collagenization, and drug-sensitive bioreactivity in healthy, metabolic dysfunction-associated steatotic liver disease, metabolic associated steato-hepatitisare not thoroughly introduced.
- In the introduction, the authors should clearly introduce the existing reviews concerning liver sinusoidal endothelial cellsand explain why the present review is important.
- In Fig. 1, the role of LSECsis not clearly illustrated.
- The source of the published pictures, such as Fig. 2, should be clearly introduced and cited.
- From section 4 to section 6, the role of LSECshas been introduced. However, the detailed mechanism are not clearly introduced and summarized through figures.
- Section 7 “3D Human Model”is not closely connected with other sections. In this section, many paragraphs lack references.
- Table 2 is not interesting.
- Conclusion should be stand alone.
Author Response
Reviewers’ Reports
REPORT 1
In this review, the authors summarize the progress on the effects of liver sinusoidal endothelial cells in regulating immunologic responses, collagenization, and drug-sensitive bioreactivity in healthy, metabolic dysfunction-associated steatotic liver disease, metabolic associated steato-hepatitis, as well as a human primary 3D model. The content of this review is useful to guide liver sinusoidal endothelial cells-related researches.
This manuscript needs some modifications:
- The tile of this review is too long, which should be refined.
Thank you for your suggestion. We shortened the title as suggested.
- Please carefully check the writing of the full names and their abbreviations, especially the capital letter and lower case letter. For example, Liver Sinusoidal Endothelial Cells (LSECs)should be written as “Liver sinusoidal endothelial cells (LSECs)” in the abstract.
Yes, thank you. We revised the manuscript for full names and abbreviation and maintained a uniformity throughout the entire manuscript.
- Abstract is confusing and not well organized, especially the impacts of liver sinusoidal endothelial cells in regulating immunologic responses, collagenization, and drug-sensitive bioreactivity in healthy, metabolic dysfunction-associated steatotic liver disease, metabolic associated steato-hepatitis are not thoroughly introduced.
Thank you. Yes, we revised the abstract and kept a uniformity into it.
- In the introduction, the authors should clearly introduce the existing reviews concerning liver sinusoidal endothelial cells and explain why the present review is important.
Thank you for this remark. This narrative review examines the distinctive functions of LSECs in regulating immunologic responses, collagenization, and drug-sensitive bioreactivity in a com-prehensive view than before due to the extensive experience of the use of electron microscopy experience for diagnosis and research. Moreover, this review highlights the LSEC influence using data of a current human primary 3D model of Metabolic Associated Steato-Hepatitis (MASH).
- In Fig. 1, the role of LSECs is not clearly illustrated.
Figure 1 was reshaped considering the role of LSECs.
- The source of the published pictures, such as Fig. 2, should be clearly introduced and cited.
Figure 2 depicts the ultrastructural examination of a pediatric patient with “CRN score” of 7 dis-playing some obliteration of fenestrae (scale bar: 1 micrometer, direct magnification 15,000 X). The CRN score refers to a scoring system developed by the NASH Clinical Research Network for assessing the severity of liver damage in patients with non-alcoholic fatty liver disease. The electron microscopic photograph belongs to the personal archive of the senior author (CMS). The photograph is fully anonymized. No copyright issue.
- From section 4 to section 6, the role of LSECs has been introduced. However, the detailed mechanisms are not clearly introduced and summarized through figures.
- Thank you for your suggestion. We made substantial changes to the text.
- Section 7 “3D Human Model”is not closely connected with other sections. In this section, many paragraphs lack references.
Thank you for your suggestion. We made substantial changes to the text.
- Table 2 is not interesting.
Thank you for your consideration. We deleted this table as suggested.
- Conclusion should be stand alone.
Thank you for your suggestion. We made a separate section.
REPORT 2
The article comprehensively explores the functions of LSECs and their involvement in liver diseases but lacks sufficient discussion on specific clinical applications and therapeutic outcomes. I have two questions: How were the LSECs specifically isolated and characterized in the study? What specific markers or methods were used to identify and track the dynamic changes of LSECs during disease progression?
Many citation markers are incorrectly placed. Line 218 has a blank gap. The font in the citation on line 222 is problematic. Figure 1 is positioned wrongly. The horizontal lines in Table 2 are formatted incorrectly. There's an extra letter "A" at position 620.
Thank you for your consideration and suggestions. We addressed the method of isolation and we expanded the therapeutic strategies in the manuscript. Moreover, we fixed the issues of the citation markers. Table 2 was deleted, because another reviewer asked us to delete it, because it was not actively contributing to the article.
REPORT 3
The article presents an informative overview of the roles of liver sinusoidal endothelial cells (LSECs) in various liver diseases, including metabolic dysfunction-associated steatotic liver disease (MASLD), metabolic-associated steatohepatitis (MASH), and cirrhosis, as well as experimental insights from a human primary 3D model. LSECs have emerged as critical players in liver homeostasis and disease progression. The manuscript is generally well-organized and makes a valuable contribution to the field of liver research. However, there are some concerns regarding the clarity of figures and the interpretation of data from in vivo models, along with some grammatical issues.
Here are some points I address as follows:
- Please carefully proofread the entire manuscript for grammar errors and inconsistencies.
Thank you for your suggestions. Yes, the manuscript was proofread for grammar errors and inconsistencies.
- The authors could use BioRender or similar software to create a visually appealing and informative illustration for Figure1 to clearly depict LSECs struction and function. Additionally, the head arrows are added in Figure2 to precisely indicate locations or processes being highlighted. The authors also could provide sufficient details to understand.
Thank you for your suggestion. Yes, figure 2 was expanded with arrows and better cartoons. We used several alternatives to BioRender software and all the software and details have been cited in the legend.
- In this manuscript, the authors acknowledge the limitations of in vivo models, particularly the differences between rodent and human LSECs. However, it doesn't consistently highlight these differences. Therefore, this manuscript needs to be more critical in evaluating and discussing the relevance of rodent studies to human liver diseases.
We expanded the manuscript with regard to the limitations of in vivo models, particularly the differences between rodent and human LSECs.
- For HCC, the authors could provide a more balanced and detailed discussion of the conflicting evidence regarding LSEC’s role in HCC development, and potential mechanism by which LSECs might either promote or inhibit HCC.
We expanded the manuscript with regard to the role of LSECs in HCC development, as requested.
Reviewer 2 Report
Comments and Suggestions for Authors
The article comprehensively explores the functions of LSECs and their involvement in liver diseases but lacks sufficient discussion on specific clinical applications and therapeutic outcomes.I have two questions: How were the LSECs specifically isolated and characterized in the study? What specific markers or methods were used to identify and track the dynamic changes of LSECs during disease progression?
Many citation markers are incorrectly placed. Line 218 has a blank gap. The font in the citation on line 222 is problematic. Figure 1 is positioned wrongly. The horizontal lines in Table 2 are formatted incorrectly. There's an extra letter "A" at position 620.
Author Response

(The authors gave the same response as above.)

Reviewer 3 Report
Comments and Suggestions for Authors
The article presents an informative overview of the roles of liver sinusoidal endothelial cells (LSECs) in various liver diseases, including metabolic dysfunction-associated steatotic liver disease (MASLD), metabolic-associated steatohepatitis (MASH), and cirrhosis, as well as experimental insights from a human primary 3D model. LSECs have emerged as critical players in liver homeostasis and disease progression. The manuscript is generally well-organized and makes a valuable contribution to the field of liver research. However, there are some concerns regarding the clarity of figures and the interpretation of data from in vivo models, along with some grammatical issues.
Here are some points I address as follows:
- Please carefully proofread the entire manuscript for grammar errors and inconsistencies.
- The authors could use BioRender or similar software to create a visually appealing and informative illustration for Figure1 to clearly depict LSECs struction and function. Additionally, the head arrows are added in Figure2 to precisely indicate locations or processes being highlighted. The authors also could provide sufficient details to understand.
- In this manuscript, the authors acknowledge the limitations of in vivo models, particularly the differences between rodent and human LSECs. However, it doesn't consistently highlight these differences. Therefore, this manuscript needs to be more critical in evaluating and discussing the relevance of rodent studies to human liver diseases.
- For HCC, the authors could provide a more balanced and detailed discussion of the conflicting evidence regarding LSEC’s role in HCC development, and potential mechanism by which LSECs might either promote or inhibit HCC.
Author Response

(The authors gave the same response as above.)

Round 2
Reviewer 1 Report
Comments and Suggestions for Authors
The manuscript has been well revised.
Reviewer 2 Report
Comments and Suggestions for Authors
No more opinions.